# Virtual Antenna Array and Fractional Fourier Transform-Based TOA Estimation for Wireless Positioning

**DOI:** 10.3390/s19030638

**Published:** 2019-02-02

**Authors:** Zhigang Chen, Lei Wang, Mengya Zhang

**Affiliations:** School of Electronic and Information Engineering, Xi’an Jiaotong University, No. 28 West Xianning Road, Xi’an 710049, China; lei.wang@mail.xjtu.edu.cn (L.W.); zhangmengya0820@163.com (M.Z.)

**Keywords:** virtual antenna array, fractional fourier transform (FRFT), time-of-arrival (TOA) estimation, channel frequency responses (CFRs), wireless positioning

## Abstract

In this paper, a novel virtual antenna array and fractional Fourier transform (FRFT)-based 2-dimension super-resolution time-of-arrival (TOA) estimation algorithm for OFDM WLAN systems has been proposed. The proposed algorithm employs channel frequency responses (CFRs) at the equi-spaced positions on a line or quasi-line moving trajectory, i.e., the CFRs of a virtual antenna array, to extract multipaths’ TOA information. Meanwhile, a new chirp-like quadratic function is used to approximate the channel multipaths’ phase variation across the space dimension, which is more reasonable than the traditional linear function, especially for relatively big virtual antenna array sizes. By exploiting the property of chirp-like multipaths’ energy concentration in the FRFT domain, the FRFT can be first used to separate chirp-like multipath components, then the existing TOA estimation methods in frequency domain can be further employed on the separated multipath components to obtain the multipaths’ TOA estimates. Therefore, the proposed algorithm can make more use of the multipaths’ characteristics in the space dimension, thus it can efficiently enhance the multipath resolution and achieve better multipaths’ TOA estimation performance without requiring a real antenna array. Simulation results demonstrate the effectiveness of the proposed algorithm.

## 1. Introduction

Indoor wireless positioning technology has increasingly attracted research interests [1,2] for its wide and popular applications in the construction industry, health industry [3], people guidance and so on. Geometric location based on time-of-arrival (TOA) is the most popular method for accurate positioning systems [4]. The basic problem in TOA estimation is to extract the TOA of the Line-of-Sight (LOS) path in wireless channels, which is usually interfered by Non-LOS (NLOS) paths in the multipath environment [5]. Due to the serious NLOS multipath effects and the limited system bandwidth, the conventional TOA estimation methods, such as the inverse Fourier transform (IFT)-based method, cannot achieve higher estimation accuracy [6].

To improve the TOA estimation accuracy, several super-resolution TOA estimation methods based on the channel frequency responses (CFRs), such as the multiple signal classification (MUSIC)-based method [1], TLS-ESPRIT [7]-based method, matrix pencil (MP)-based method [8,9] and so on, employ the eigenstructure or eigen-subspace of the CFRs to obtain TOA estimates. In contrast to the conventional methods, these super-resolution methods exploit not only the property of multipaths’ linear phase variation across the frequency dimension but also the sparsity of the main channel multipaths in the time-delay domain, so they can achieve higher accuracy than the conventional methods. Nevertheless, such super-resolution methods are susceptible to the noise since the noise causes serious estimation errors of eigen-subspace or eigenvalues.

Furthermore, the 2-dimension super-resolution TOA estimation methods, such as the 2-dimension MP method [10], joint beamforming and MUSIC-based method [11], and the 2-dimension (2D) MUSIC method [12], employ both the channel frequency responses and the channel space responses to jointly estimate the TOA and AOA. Besides the property of multipaths’ linear phase differences across the frequency dimension and the sparsity of the main channel multipaths, the 2D super-resolution methods also exploits the property of multipaths’ quasi-linear phase variation across the space dimension, hence they can efficiently mitigate the noise effects and further improve the TOA estimation performance. However, such 2D super-resolution methods are limited by the requirement of an antenna array to obtain the channel space responses, and the assumption of quasi-linear phase variation property across the space dimension, which implies that the antenna array size should be small enough.

Without requiring an antenna array and suffering from the limitation of its relatively small size, a novel virtual antenna array and fractional Fourier transform (FRFT)-based 2D super-resolution TOA estimation method has been proposed in this paper. By using the built-in Kinect or inertial sensors in the mobile terminal (MT), the proposed method first obtains the CFRs at the equi-spaced positions on a line or quasi-line moving trajectory as the CFRs of a virtual antenna array. Meanwhile, considering that the size of virtual antenna array is not limited by the hardware and not precisely known, the channel multipaths’ phase variation across the space dimension is approximated by a chirp-like quadratic function, which is more reasonable than a traditional linear function. Based on the CFRs of a virtual antenna array, the proposed method employs the FRFT [13] to separate and extract multipath components in CFRs by exploiting the property of the chirp’s energy concentration in the FRFT domain [14], then it uses the existing time-delay estimation methods, such as the IFT method and the MP method [9], to estimate the TOAs of the separated multipaths. Therefore, by employing the virtual antenna array idea, modeling the multipath phase variation as a quadratic function, the proposed FRFT-based algorithm can make more use of the multipaths’ characteristics in the space domain and achieve more robust TOA estimation performance without requiring an antenna array.

The main contributions of this paper are as follows: (1) We present a novel virtual antenna array system model for positioning by regarding the CFRs at the equi-spaced positions on a line or quasi-line moving trajectory, which can be obtained by using built-in Kinect or inertial sensors in the MT, as CFRs of a virtual antenna array. (2) The channel multipaths’ phase variation across the space dimension is modeled as a chirp-like quadratic function instead of a traditional linear function. Due to being high-order approximation, the chirp-like quadratic function suffers from much less multipath phase model mismatch error for the unlimited and unspecified size of virtual antenna arrays. (3) An FRFT-based 2D super-resolution TOA estimation method has been proposed. By exploiting the chirp’s energy concentration property in the FRFT domain, the FRFT is employed to separate chirp-like multipath components in the space domain, then the existing TOA estimation methods, such as the IFT method and the MP method [9], are used to obtain multipaths’ TOA estimate. Thus, the FRFT-IFT algorithm and the FRFT-MP algorithm have been developed. (4) Simulation results demonstrate that the proposed method can achieve more robust TOA estimation performance than the 2D MP super-resolution TOA estimation method due to making more of the multipaths’ characteristics in the space domain.

The rest of the paper is organized as follows. In Section 2, the virtual antenna array system model is described. In Section 3, the properties of multipath components in the CFRs of a virtual antenna array will be analyzed first by FRFT, and a novel FRFT-based TOA estimation algorithm will be then developed based on these properties. Simulation results are presented in Section 4 to evaluate the performance of the proposed method. Finally, conclusions are drawn in Section 5.

## 2. Virtual Antenna Array System Model

For ease of exposition, we consider a basic OFDM WLAN system with single AP (Access Point) and single MT. The following system model can be extended directly to the WLAN system with multiple APs and multiple MTs. Suppose that the MT moves in a straight-line or quasi-straight-line during a short period, which is usually the case with pedestrians‘ walking way in indoor environments. If the MT’s acceleration, which can be measured by built-in Kinect or inertial sensors in the MT [15], is smaller than a chosen threshold, the MT can be assumed to make uniform linear motion with its speed imprecisely estimated due to the accumulated error. By simply taking positions at *M* equi-interval moments during such short period of relatively small acceleration or even employing the pedestrian dead reckoning method [16,17], we can obtain *M* equi-spaced positions on a straight-line or quasi-straight-line moving trajectory of the MT and regard the CFRs at such positions as the CFRs of a virtual antenna array. Suppose the central position of the trajectory as the position of interest and call the CFRs at the *M* positions on the trajectory as the CFRs of a virtual antenna array at this target position. Denote the distance between adjacent positions as ρ, i.e., the virtual antenna array element spacing, which is assumed constant but not precisely estimated due to the imprecisely estimated speed and the negligible incremental accumulated error in a short period. The channel impulse response (CIR) between the AP and the MT at the *m*-th position of the moving trajectory is given by [9]
(1)$$h\left(t,m\right)=\sum_{l=1}^{L}b_{l}a\left(\tau_{m,l}\right)\delta \left(t-\tau_{m,l}\right)\;\;\;\;m=-\frac{M}{2},\cdots ,0,\cdots ,\frac{M}{2}-1$$
where *L* is the number of distinct propagation paths, $b_{l}$ and $\tau_{m,l}$ are the complex gain and the TOA of the *l*-th path, respectively, $a\left(\tau_{m,l}\right)=e^{-j2\pi f_{c}\tau_{m,l}}$ is the array response of the mth antenna to the *l*-th path.

Assume the direction of arrival (DOA) relative to the moving trajectory of the *l*th path at the central position, i.e., the 0-th position on the trajectory as $\theta_{l}$, which is shown in Figure 1, we have
(2)$$\begin{array}{c} \tau_{m,l}=\frac{1}{c}\sqrt{{\left(m\rho \right)}^{2}+{\left(c\tau_{0,l}\right)}^{2}-2cos\theta_{l}\left(m\rho \right)\left(c\tau_{0,l}\right)}\\ \;\approx \frac{1}{c}\left(c\tau_{0,l}-\left(m\rho \right)cos\theta_{l}+\frac{1}{2}\frac{{\left(m\rho sin\theta_{l}\right)}^{2}}{c\tau_{0,l}}\right)\;\;\;\;m=-\frac{M}{2},\cdots ,0,\cdots ,\frac{M}{2}-1 \end{array}$$
where *c* is the speed of light, the approximation holds if the virtual antenna array size Mρ is deliberately chosen to satisfy $\frac{M\rho }{2c\tau_{0,l}}\ll 1$. This approximation is obtained by using second order Taylor series expansion, thus it is more reasonable than the conventional linear function, i.e., the first order Taylor series expansion of $\tau_{m,l}$, considering the size of the virtual antenna array is unspecified and not limited by the hardware.

From (1) and (2), the estimated CFR for the *n*th subcarrier and the *m*-th position on the moving trajectory at the receiver side can be expressed as
(3)$$\begin{array}{cc} {\hat{H}}_{m,n} & =H_{m,n}+W_{m,n}\\ & =\sum_{l=1}^{L}b_{l}e^{-j\omega_{c}\tau_{m,l}}e^{-j2\pi \Delta f\tau_{m,l}}+W_{m,n}\\ & \approx \sum_{l=1}^{L}\{b_{l}e^{-j\omega_{c}\left(\tau_{0,l}-m\rho cos\theta_{l}/c+{\left(m\rho sin\theta_{l}\right)}^{2}/\left(2\tau_{0,l}c^{2}\right)\right)}\cdot \\ & e^{-j2\pi n\Delta f(\tau_{0,l}-m\rho cos\theta_{l}/c+{\left(m\rho sin\theta_{l}\right)}^{2}/\left(2\tau_{0,l}c^{2}\right))}\}+W_{m,n}\\ & \;\;\;\;\;m=-\frac{M}{2},\cdots ,\frac{M}{2}-1;n=0,1,\cdots ,N \end{array}$$
as in [10], where $\omega_{c}=2\pi f_{c}$ is the carrier angular frequency, Δf, *N* and $W_{m,n}$ denote the OFDM subcarrier spacing, the subcarrier number and the additive white Gaussian noise (AWGN), respectively. For simplicity, the CFR can be also written as
(4)$$\begin{array}{cc} {\hat{H}}_{m,n} & \approx \sum_{l=1}^{L}\left\{b_{l}e^{j\varphi_{l}\left(n\right)+j2\pi f_{l}\left(n\right)m\rho +j\pi \mu_{l}\left(n\right){\left(m\rho \right)}^{2}}\right\}+W_{m,n} \end{array}$$
where
(5)$$\begin{array}{c} \varphi_{l}\left(n\right)=-\left(\omega_{c}+2\pi n\Delta f\right)\tau_{0,l}\\ f_{l}\left(n\right)=\left(f_{c}+n\Delta f\right)cos\theta_{l}/c\\ \mu_{l}\left(n\right)=-\left(f_{c}+n\Delta f\right){\left(sin\theta_{l}\right)}^{2}/\left(\tau_{0,l}c^{2}\right) \end{array}$$

## 3. Fractional Fourier Transform Based TOA Estimation

It can be observed from (4) that the phases of multipath components in CFRs ${\left\{H_{m,n}\right\}}_{m=1,n=1}^{M,N}$ across the frequency dimension vary linearly, while the phase variations of multipath components in CFRs across the space dimension approximate quadratic functions, in other words, the CFRs over the space dimension can be equivalently viewed as *L* superimposed chirp-like multipath components. Consequently, the FRFT in the space domain and the frequency domain processing method, such as the IFT or MP method, can be jointly employed to extract multipath components’ parameters.

In this section, the multipath components of the CFRs will be analyzed first by employing FRFT across the space dimension, and a novel FRFT-based multipath TOA estimation algorithm will be then developed.

### 3.1. Properties of Multipath Components in FRFT Domain

To separate and extract the chirp-like components, a typical discrete fractional Fourier transform (DFRFT) [18] will be implemented on the CFRs at each subcarrier across the space dimension as
(6)$$\begin{array}{c} {\hat{H}}_{n}\left(\alpha ,u\right)=F^{\alpha }\left({\left\{H_{m,n}\right\}}_{m=-\frac{M}{2}}^{\frac{M}{2}-1}\right)+F^{\alpha }\left({\left\{W_{m,n}\right\}}_{m=-\frac{M}{2}}^{\frac{M}{2}-1}\right)\\ =\frac{A_{\alpha }}{\sqrt{M}}e^{j\pi u^{2}cot\alpha }\sum_{m=-\frac{M}{2}}^{\frac{M}{2}-1}\left\{e^{-j2\pi ucsc\left(\alpha \right)/\sqrt{M}}e^{j\pi u^{2}cot\left(\alpha \right)/M}H_{m,n}\right\}\\ +W_{n}\left(\alpha ,u\right)\;\;\;\;\;n=0,1,\cdots ,N \end{array}$$
where α=pπ/2 and *p* is the FRFT order, $A_{\alpha }=\sqrt{\frac{1-jcot(\alpha )}{2\pi }}$, $F^{\alpha }\left(•\right)$ denotes the DFRFT operation of order α=pπ/2, $W_{n}\left(\alpha ,u\right)$ is the transformed AWGN by DFRFT.

By substituting (4) into the first part of (6), the noise-free transformed CFRs by DFRFT can be expressed as
(7)$$\begin{array}{c} H_{n}\left(\alpha ,u\right)\approx \frac{A_{\alpha }}{\sqrt{M}}e^{j\pi u^{2}cot\alpha }\sum_{m=-\frac{M}{2}}^{\frac{M}{2}-1}\{e^{-j2\pi umcsc\left(\alpha \right)/\sqrt{M}}e^{j\pi m^{2}cot\left(\alpha \right)/M}\cdot \\ \sum_{l=1}^{L}\left\{b_{l}e^{j\varphi_{l}\left(n\right)+j2\pi f_{l}\left(n\right)m\rho +j\pi \mu_{l}\left(n\right){\left(m\rho \right)}^{2}}\right\}\}\\ =\frac{A_{\alpha }}{\sqrt{M}}e^{j\pi u^{2}cot\alpha }\sum_{l=1}^{L}\{b_{l}e^{j\varphi_{l}\left(n\right)}\sum_{m=-\frac{M}{2}}^{\frac{M}{2}-1}\{e^{j2\pi [-ucsc\left(\alpha \right)/\sqrt{M}+f_{l}\left(n\right)\rho ]m}\cdot \\ e^{j\pi \left[cot\left(\alpha \right)/M+\mu_{l}\left(n\right)\rho^{2}\right]m^{2}}\}\}n=0,1,\cdots ,N \end{array}$$

Define the DFRFT signal in (7) contributed only by the *l*-th multipath component as $S_{l,n}\left(\alpha ,u\right)$
(8)$$\begin{array}{c} S_{l,n}\left(\alpha ,u\right)=c_{l}e^{j\varphi_{l}\left(n\right)}\sum_{m=-\frac{M}{2}}^{\frac{M}{2}-1}\left\{e^{j2\pi [-ucsc\left(\alpha \right)/\sqrt{M}+f_{l}\left(n\right)\rho ]m}\cdot e^{j\pi \left[cot\left(\alpha \right)/M+\mu_{l}\left(n\right)\rho^{2}\right]m^{2}}\right\}\\ n=0,1,\cdots ,N \end{array}$$
where
$$\begin{array}{c} c_{l}=\frac{A_{\alpha }}{\sqrt{M}}e^{j\pi u^{2}cot\alpha }b_{l} \end{array}$$

Then it can be derived that $S_{l,n}\left(\alpha ,u\right)$ possesses the following properties.

**Property** **1.**
*Due to the chirp’s energy concentration property in the DFRFT domain [14], $S_{l,n}\left(\alpha ,u\right)$ has a distinguishable peak in the (α,u) plane, the peak, and its coordinate $\left({\overset{´}{\alpha }}_{l}\left(n\right),{\overset{´}{u}}_{l}\left(n\right)\right)$ correspond to*
(9)$$\begin{array}{cc} S_{l,n}\left({\overset{´}{\alpha }}_{l}\left(n\right),{\overset{´}{u}}_{l}\left(n\right)\right) & \approx A_{{\overset{´}{\alpha }}_{l}\left(n\right)}\sqrt{M}e^{j\pi {\left({\overset{´}{u}}_{l}\left(n\right)\right)}^{2}cot\left({\overset{´}{\alpha }}_{l}\left(n\right)\right)}b_{l}e^{j\varphi_{l}\left(n\right)}\\ {\overset{´}{\alpha }}_{l}\left(n\right) & =arccot(-\mu_{l}\left(n\right)M\rho^{2})\\ {\overset{´}{u}}_{l}\left(n\right) & =f_{l}\left(n\right)\sqrt{M}\rho /csc\left({\hat{\alpha }}_{l}\left(n\right)\right) \end{array}$$


**Property** **2.**
*It can be proved that $\sum_{n=1}^{N}{\left|S_{l,n}\left(\alpha ,u\right)\right|}^{2}$ achieves its maximum value at $\left({\overset{´}{\alpha }}_{l}\left(\frac{N}{2}\right),{\overset{´}{u}}_{l}\left(\frac{N}{2}\right)\right)$, i.e.,*
(10)$$\begin{array}{c} \left({\overset{´}{\alpha }}_{l}\left(\frac{N}{2}\right),{\overset{´}{u}}_{l}\left(\frac{N}{2}\right)\right)\approx \arg\max_{\left(\alpha ,u\right)}\left\{\sum_{n=1}^{N}{\left|S_{l,n}\left(\alpha ,u\right)\right|}^{2}\right\} \end{array}$$
*and decreases rapidly with $|\alpha -{\overset{´}{\alpha }}_{l}\left(N/2\right)|$ or $|u-{\overset{´}{u}}_{l}\left(N/2\right)|$ increasing (See Appendix A for the derivation).*


**Property** **3.**
*It can also be proved that if the Euclidean distance between any two of ${\left\{\left({\overset{´}{\alpha }}_{l}\left(\frac{N}{2}\right),{\overset{´}{u}}_{l}\left(\frac{N}{2}\right)\right)\right\}}_{l=1}^{L}$ is sufficiently large, $\sum_{n=1}^{N}|H_{n}{\left(\alpha ,u\right)|}^{2}=\sum_{n=1}^{N}{\left|\sum_{l=1}^{L}S_{l,n}\left(\alpha ,u\right)\right|}^{2}$ has L distinguishable peaks corresponding to the L multipath components in the (α,u) plane and the peaks’ coordinates correspond to ${\left\{\left({\overset{´}{\alpha }}_{l}\left(\frac{N}{2}\right),{\overset{´}{u}}_{l}\left(\frac{N}{2}\right)\right)\right\}}_{l=1}^{L}$ (See Appendix B for the derivation).*


### 3.2. DFRFT Based Multipath TOA Estimation

Based on the above properties of multipath components in (8), the parameters of multipath components can be estimated through searching for the peaks of CFRs in the FRFT domain. However, the estimation error caused by interferences and/or AWGN cannot be neglected in cases of insufficiently large virtual antenna size M×ρ or at low SNRs.

Due to the Property 3 above, an averaging method over *N* subcarriers can be used to reduce the estimation error caused by noise. Furthermore, a successive interference cancellation (SIC) method can be employed to mitigate the interference effects as in [19]. Thus, the parameters of multipath components will be obtained by combining the SIC method and the averaging method.

Without loss of generality, we assume that the amplitudes $\left\{\right|b_{l}{\left|\right\}}_{l=1}^{L}$ of *L* paths in the channel satisfy $|b_{1}\left|>\right|b_{2}\left|\cdots \right|b_{L}|$. Considering the SIC and averaging-based multipath parameters estimation is composed of $L^{\prime }$ similar iterative steps, only the steps in the $l^{\prime }$-th iteration are exemplified as follows. Assume the initial filtered CFRs ${\tilde{H}}_{m,n}^{\left(0\right)}={\hat{H}}_{m,n}$.

#### 3.2.1. Searching for the Peak

Given the filtered CFRs ${\tilde{H}}_{m,n}^{\left(l^{\prime }-1\right)}$ in the $\left(l^{\prime }-1\right)$-th step, the peak of $\sum_{n=1}^{N}{\left|{\tilde{H}}_{n}^{\left(l^{\prime }-1\right)}\left(\alpha ,u\right)\right|}^{2}$ in the (α,u) plane at the $l^{\prime }$-th iteration can be expressed as
(11)$$\begin{array}{c} \left({\hat{\alpha }}^{\left(l^{\prime }\right)},{\hat{u}}^{\left(l^{\prime }\right)}\right)=\arg\max_{\left(\alpha ,u\right)}\left\{\sum_{n=1}^{N}{\left|{\tilde{H}}_{n}^{\left(l^{\prime }-1\right)}\left(\alpha ,u\right)\right|}^{2}\right\}\\ =\arg\max_{\left(\alpha ,u\right)}\left\{\sum_{n=1}^{N}{\left|F^{\alpha }\left({\left\{{\tilde{H}}_{m,n}^{\left(l^{\prime }-1\right)}\right\}}_{m=-\frac{M}{2}}^{\frac{M}{2}-1}\right)\right|}^{2}\right\} \end{array}$$

The estimated peak corresponds to the strongest multipath component in the filtered CFRs ${\tilde{H}}_{m,n}^{\left(l^{\prime }-1\right)}$, i.e., the $l^{\prime }$-th strongest multipath component in the CFRs ${\left\{{\hat{H}}_{m,n}\right\}}_{m=-\frac{M}{2}}^{\frac{M}{2}-1}$. According to the Property 3 above, it approximates the peak of the DFRFT of the $l^{\prime }$-th multipath component at the N/2-th subcarrier in the (α,u) plane.
(12)$$\begin{array}{c} \left({\hat{\alpha }}^{\left(l^{\prime }\right)},{\hat{u}}^{\left(l^{\prime }\right)}\right)\approx \left({\overset{´}{\alpha }}_{l^{\prime }}\left(\frac{N}{2}\right),{\overset{´}{u}}_{l^{\prime }}\left(\frac{N}{2}\right)\right) \end{array}$$

Since the DFRFT transformed signal for each subcarrier means averaging the noise over *M* positions on the trajectory and the summation averages the noise energy over *N* subcarriers, $\sum_{n=1}^{N}{\left|{\tilde{H}}_{n}^{\left(l^{\prime }-1\right)}\left(\alpha ,u\right)\right|}^{2}$ contains the averaged noise energy across both the frequency and space dimensions, thus the noise effects on the estimation $\left({\hat{\alpha }}^{\left(l^{\prime }\right)},{\hat{u}}^{\left(l^{\prime }\right)}\right)$ can be reduced greatly considering that the subcarrier number and the number of the virtual antenna array elements are usually large.

From (5) and (9), the peak of the DFRFT of the $l^{\prime }$-th multipath component at the *n*-th subcarrier in the (α,u) plane can be derived as
(13)$$\begin{array}{c} {\hat{\overset{´}{\alpha }}}_{l^{\prime }}\left(n\right)=arccot\left(\frac{f_{c}+n\Delta f}{f_{c}+N\Delta f/2}cot\left({\hat{\alpha }}^{\left(l^{\prime }\right)}\right)\right)\\ {\hat{\overset{´}{u}}}_{l^{\prime }}\left(n\right)=\frac{f_{c}+n\Delta f}{f_{c}+N\Delta f/2}\frac{csc({\hat{\alpha }}^{\left(l^{\prime }\right)})}{csc({\hat{\overset{´}{\alpha }}}_{l^{\prime }}\left(n\right))}{\hat{u}}^{\left(l^{\prime }\right)} \end{array}$$

#### 3.2.2. Estimating the TOA of the Strongest Multipath

From the relationship between the peaks of the DFRFT of the CFRs and the parameters of *L* multipath-chirp components in (7), the peak of the DFRFT at the *n*-th subcarrier is compensated as
(14)$$\begin{array}{c} {\overset{´}{\tilde{H}}}_{n}^{\left(l^{\prime }-1\right)}\left({\hat{\overset{´}{\alpha }}}_{l^{\prime }}\left(n\right),{\hat{\overset{´}{u}}}_{l^{\prime }}\left(n\right)\right)\\ ={\tilde{H}}_{n}^{\left(l^{\prime }-1\right)}\left({\hat{\overset{´}{\alpha }}}_{l^{\prime }}\left(n\right),{\hat{\overset{´}{u}}}_{l^{\prime }}\left(n\right)\right)A_{{\hat{\overset{´}{\alpha }}}_{l^{\prime }}\left(n\right)}^{*}e^{-j\pi {\left({\hat{\overset{´}{u}}}_{l^{\prime }}\left(n\right)\right)}^{2}cot{\hat{\overset{´}{\alpha }}}_{l^{\prime }}\left(n\right)}\\ \approx |A_{{\hat{\overset{´}{\alpha }}}_{l^{\prime }}\left(n\right)}{|}^{2}\sqrt{M}b_{l^{\prime }}e^{j\varphi_{l^{\prime }}\left(n\right)}+{\overset{´}{\tilde{W}}}_{n}\left({\hat{\overset{´}{\alpha }}}_{l^{\prime }}\left(n\right),{\hat{\overset{´}{u}}}_{l^{\prime }}\left(n\right)\right)+\\ \underset{I_{n}}{\underbrace{\left\{\sum_{l=l^{\prime }+1}^{L}S_{l,n}\left({\hat{\overset{´}{\alpha }}}_{l^{\prime }}\left(n\right),{\hat{\overset{´}{u}}}_{l^{\prime }}\left(n\right)\right)\right\}\cdot A_{{\hat{\overset{´}{\alpha }}}_{l^{\prime }}\left(n\right)}^{*}e^{-j\pi {\left({\hat{\overset{´}{u}}}_{l^{\prime }}\left(n\right)\right)}^{2}cot{\hat{\overset{´}{\alpha }}}_{l^{\prime }}\left(n\right)}}} \end{array}$$
where the approximation holds since the first $l^{\prime }-1$ strongest multipath component are almost removed in the previous $l^{\prime }-1$ iterations, the first term in the right hand represents the compensated DFRFT of the strongest multipath component, ${\overset{´}{\tilde{W}}}_{n}\left({\hat{\overset{´}{\alpha }}}_{l^{\prime }}\left(n\right),{\hat{\overset{´}{u}}}_{l^{\prime }}\left(n\right)\right)$ and $I_{n}$ are the compensated and filtered AWGN and interferences caused by the remaining multipath components, respectively.

By substituting (5) into (14), the compensated DFRFT at the peak $\left({\hat{\overset{´}{\alpha }}}_{l^{\prime }}\left(n\right),{\hat{\overset{´}{u}}}_{l^{\prime }}\left(n\right)\right)$ in (14) can be further expressed as
(15)$$\begin{array}{c} {\overset{´}{\tilde{H}}}_{n}^{\left(l^{\prime }-1\right)}\left({\hat{\overset{´}{\alpha }}}_{l^{\prime }}\left(n\right),{\hat{\overset{´}{u}}}_{l^{\prime }}\left(n\right)\right)\\ \approx d_{l^{\prime }}e^{-j2\pi n\Delta f\tau_{0,l^{\prime }}}+{\overset{´}{\tilde{W}}}_{n}\left({\hat{\overset{´}{\alpha }}}_{l^{\prime }}\left(n\right),{\hat{\overset{´}{u}}}_{l^{\prime }}\left(n\right)\right)+I_{n} \end{array}$$
with
(16)$$\begin{array}{c} d_{l^{\prime }}={\left|A_{{\hat{\overset{´}{\alpha }}}_{l^{\prime }}\left(n\right)}\right|}^{2}\sqrt{M}b_{l^{\prime }}e^{-j\omega_{c}\tau_{0,l^{\prime }}} \end{array}$$

From (15), ${\overset{´}{\tilde{H}}}_{n}^{\left(l^{\prime }-1\right)}\left({\hat{\overset{´}{\alpha }}}_{l^{\prime }}\left(n\right),{\hat{\overset{´}{u}}}_{l^{\prime }}\left(n\right)\right)$ can be viewed as a new channel frequency response at the *n*-th subcarrier. Due to the energy concentration property of DFRFT and the uniformly distributed noise in the DFRFT domain, the strongest multipath component dominates the compensated DFRFT of the *n*-th subcarrier at the peak $\left({\hat{\overset{´}{\alpha }}}_{l^{\prime }}\left(n\right),{\hat{\overset{´}{u}}}_{l^{\prime }}\left(n\right)\right)$. Hence, this new channel has a dominated multipath of interest, which has the same TOA $\tau_{0,l^{\prime }}$ as the $l^{\prime }$ multipath.

Therefore, the existing TOA estimation methods, such as the IFT method or the MP-based method [9], can then be directly used on ${\left\{{\overset{´}{\tilde{H}}}_{n}^{\left(l^{\prime }-1\right)}\left({\hat{\overset{´}{\alpha }}}_{l^{\prime }}\left(n\right),{\hat{\overset{´}{u}}}_{l^{\prime }}\left(n\right)\right)\right\}}_{n=1}^{N}$ to obtain the estimate of the strongest multipath’s TOA in this iteration, which is denoted as ${\hat{\tau }}_{0,l^{\prime }}$. Moreover, the estimate of the strongest multipath’s TOA is irrelevant to the virtual antenna array element spacing ρ from (16). It is worth mentioning, although the MP-based TOA estimation is susceptible to the noise, it can remove most interference in such new equivalent channel, especially at high SNRs.

#### 3.2.3. Cancelling the Strongest Component

Furthermore, the filtered CFRs ${\tilde{H}}_{m,n}^{\left(l^{\prime }-1\right)}$ at the *n*-th subcarrier in the $\left(l^{\prime }-1\right)$-th step will be transformed by DFRFT of order ${\hat{\overset{´}{\alpha }}}_{l^{\prime }}\left(n\right)$ as
(17)$$\begin{array}{c} {\tilde{H}}_{n}^{\left(l^{\prime }-1\right)}\left({\hat{\overset{´}{\alpha }}}_{l^{\prime }}\left(n\right),u\right)=F^{{\hat{\overset{´}{\alpha }}}_{l^{\prime }}\left(n\right)}\left({\left\{{\tilde{H}}_{m,n}^{\left(l^{\prime }-1\right)}\right\}}_{m=1}^{M}\right)\;\;\;n=0,1,\cdots ,N \end{array}$$

For each *u* in a neighborhood of ${\hat{\overset{´}{u}}}_{l^{\prime }}\left(N/2\right)$, $\left[{\tilde{H}}_{0}^{\left(l^{\prime }-1\right)}\left({\hat{\overset{´}{\alpha }}}_{l^{\prime }}\left(0\right),u\right),\cdots ,{\tilde{H}}_{N-1}^{\left(l^{\prime }-1\right)}\left({\hat{\overset{´}{\alpha }}}_{l^{\prime }}\left(N-1\right),u\right)\right]$ is formed as a new equivalent CFR vector over *N* subcarriers. Similarly, it can be derived from (7) that the $l^{\prime }$-th multipath component is dominant in such formed CFR vector. Again, the IFT or MP-based multipath parameters estimation method [9] can be employed on such a CFR vector to obtain the TOA and the complex gain of its strongest multipath component, which are denoted as ${\tilde{\tau }}_{0,l^{\prime }}\left(u\right)$ and ${\tilde{b}}_{0,l^{\prime }}\left(u\right)$. Consequently, the DFRFT at *u* can be filtered by cancelling this strongest multipath component from such a CFR vector as
(18)$$\begin{array}{c} {\tilde{H}}_{n}^{\left(l^{\prime }\right)}\left({\hat{\overset{´}{\alpha }}}_{l^{\prime }}\left(n\right),u\right)\\ ={\tilde{H}}_{n}^{\left(l^{\prime }-1\right)}\left({\hat{\overset{´}{\alpha }}}_{l^{\prime }}\left(n\right),u\right)-{\tilde{b}}_{0,l^{\prime }}\left(u\right)*e^{-j2\pi n\Delta f{\tilde{\tau }}_{0,l^{\prime }}\left(u\right)}\\ \;\;\;n=0,1,\cdots ,N \end{array}$$
if the TOA estimate difference $|{\tilde{\tau }}_{0,l^{\prime }}\left(u\right)-{\hat{\tau }}_{0,l^{\prime }}|$ is smaller than a chosen threshold.

Then the filtered DFRFT ${\tilde{H}}_{n}^{\left(t\right)}\left({\hat{\overset{´}{\alpha }}}_{l^{\prime }}\left(n\right),u\right)$ is transformed back to the frequency domain by DFRFT of order $-{\hat{\overset{´}{\alpha }}}_{l^{\prime }}\left(n\right)$. Hence, the filtered CFRs in the $l^{\prime }$-th iteration can be expressed as
(19)$$\begin{array}{c} {\tilde{H}}_{m,n}^{\left(l^{\prime }\right)}=F^{-{\hat{\overset{´}{\alpha }}}_{l^{\prime }}\left(n\right)}\left({\tilde{H}}_{n}^{\left(t\right)}\left({\hat{\overset{´}{\alpha }}}_{l^{\prime }}\left(n\right),u\right)\right)\;\;\;n=0,1,\cdots ,N \end{array}$$

The above procedures will repeat until the amplitude of the detected multipath component is smaller than a certain threshold. Finally, the TOA of the direct multipath can be estimated as the minimum one among ${\left\{{\hat{\tau }}_{0,l^{\prime }}\right\}}_{l^{\prime }=1}^{L^{\prime }}$.

## 4. Simulation Results and Discussions

### 4.1. Simulation Settings

To evaluate the performance of the proposed algorithm, we have conducted computer simulations in a relatively simple 9 m × 7 m × 4 m (length × width × height) three-dimension indoor WLAN environment, in which there are 1 AP located at (0.5,4.5,2.5) and four mixed cement and glass walls. The 2-dimension layout of the indoor WLAN environment is shown in Figure 2. The classical ray-tracing propagation model [20] is adopted in the simulations to generate the CIRs at indoor positions, and the dielectric constant and electrical conductivity of concrete and glass are respectively $\epsilon_{r1}=8,\sigma_{1}=0.01$ and $\epsilon_{r2}=2,\sigma_{2}=0.001$, and the system measurement noise is Gaussian. For simplicity, only the indirect paths created by a single specular reflection from a side wall and the direct path are considered in the simulations. The carrier frequency, subcarrier number, and subcarrier spacing are assumed as 5.0 GHz, 64, and 625 KHz, respectively.

Two kinds of scenarios have been studied in the simulations: LOS scenarios and NLOS scenarios. In the LOS scenarios, all the multipaths’ gain are generated according to the ray-tracing propagation model, thus the LOS multipath is the strongest one among all the multipaths. In the NLOS scenarios, all multipaths except the LOS multipath have the same gain as in LOS scenarios, while the LOS path is additionally attenuated by 15 dB to simulate the presence of human body or some barrier between the AP and MT according to [21,22].

To average the LOS path TOA estimation performance over space, the target positions, i.e., the central positions of MT’s moving trajectories, are set as uniformly spaced positions with the distance of 1.5 m between neighboring central positions in this indoor environment, thus there are 20 uniformly spaced target positions also shown in Figure 2. Based on the CFRs at the positions equi-spaced on a line trajectory, the direct path’s TOA at the central position of the corresponding line trajectory is obtained as the target TOA estimate in the simulations. The mean square error (MSE) of the LOS path’s TOA estimate (unit: ns) is employed as the performance measurement and all results are obtained by averaging 400 independent simulations: 20 independent simulations at each one of 20 uniformly spaced target positions.

In the simulations, we compare the performance of the proposed algorithms, the FRFT-IFT algorithm and the FRFT-MP algorithm, with the 2D MP-based super-resolution TOA estimation algorithm [10]. In each iteration of the proposed algorithms, the DFRFT is implemented on the CFRs across the space domain at each subcarrier according to (6) with ${\left\{\alpha =p\pi /2,p=q/800\right\}}_{q=-800}^{800}$, ${\left\{u=m/\sqrt{M}\right\}}_{m=-\frac{M}{2}}^{\frac{M}{2}-1}$ and M=60, and the parameters of the strongest multipath components is extracted by searching the peak in the discretized FRFT domain according to (11), then the FRFT-IFT algorithm and the FRFT-MP algorithm respectively employ the IFT scheme and MP scheme [9] to estimate the strongest multipath’s TOA, the neighborhood of the peak in the *u* domain is set as the whole *u* domain and the TOA estimate difference threshold is set as 1ns in the step of cancelling the strongest component. The iteration repeats until the energy of the extracted component is 20 times less than that in the first iteration.

### 4.2. Simulation Results

It is worth noting that the performance of the three algorithms are mainly dependent on the noise, the multipath components interference and the mismatch errors of the multipaths’ phase model. In other words, the performance of the LOS path TOA estimation depends on the ‘SINR’ of the LOS path component, i.e., the ratio of the LOS path energy to the sum of the energy of noise, the multipath components interference and the mismatch errors. Since the 2D MP method almost removes the multipath interference in both frequency and space domain, the performance of the 2D MP TOA estimation algorithm is mainly relevant to the noise and the multipaths’ phase model mismatch errors. Due to adopting the quadratic function as the multipath phase variation model across the space dimension, the two proposed algorithms: the FRFT-IFT algorithm and the FRFT-MP algorithm, suffer from much less multipaths’ phase model mismatch errors. Besides the mismatch errors and the noise, the FRFT-IFT algorithm is also subject to multipath components interference caused by the insufficient virtual antenna array size, while the FRFT-MP method can remove most of the multipath interference by employing MP-based TOA estimation method [9] in frequency domain, thus it is only subject to the residual multipath components interference. Moreover, the ‘SNR’ in the following simulations means the ratio of the sum of all the multipaths’ energy to the noise energy, whereas the ‘SNR of the LOS path component’ means the LOS path energy to the noise energy.

Figure 3 demonstrates the MSE of the TOA estimation versus SNR in LOS scenarios and NLOS scenarios, respectively. In the simulations, the virtual antenna array size are assumed as Mρ = 60 × 2 cm. From Figure 3, it is obvious that the proposed algorithms: the FRFT-MP algorithm and the FRFT-IFT algorithm generally outperform the 2D MP algorithm since the proposed algorithms can make more of the multipaths’ characteristics in the space domain. Figure 3a shows that the two proposed algorithms perform better than the 2D MP algorithm in NLOS scenarios, especially at low SNRs, because in such scenarios the noise dominates the MSE performance at low SNRs, and the proposed methods concentrate the LOS multipath energy by FRFT, thus can achieve higher ‘SINR’ of the LOS path component than the 2D MP method, while the 2D MP method suffers from noise threshold effects [23].

In LOS scenarios and high SNR cases in NLOS scenarios, the 2D MP algorithm and the FRFT-IFT algorithm are mainly subject to the mismatch error of the quasi-linear phase model and the multipath components’ interference, respectively, since the SNR of the LOS path component is sufficiently high in such cases. This explains that both these two methods have error floor effects at high SNRs in NLOS scenarios as shown in Figure 3a and their performance improves slightly with SNR increasing in LOS scenarios as shown in Figure 3b. The FRFT-MP method can mitigate the multipath interference by employing MP-based TOA estimation method [9], thus the FRFT-MP method has the lower error floor as shown in Figure 3a. On the other hand, since the MP scheme is susceptible to the noise, the FRFT-MP method performs worse than the FRFT-IFT method at low SNRs in both NLOS and LOS scenarios, as shown in Figure 3a,b.

Figure 4 demonstrates the MSE of the TOA estimation versus different sizes of virtual antenna array at SNR = 25 dB. Please note that the SNR of the LOS path component is not very high due to the relatively weak LOS path component in NLOS scenarios. As the virtual antenna array size increases, more multipaths’ characteristics in the space domain can be exploited to enhance the ‘SINR’ of the LOS path component until the multipaths’ phase model mismatch errors are not negligible at cases of relatively big virtual antenna array size in NLOS scenarios. This explains that the performance of the three algorithms improve with the virtual antenna array size increasing from 60 × 0.5 cm to 60 × 2.5 cm, but they are worsened at the case of virtual antenna array size 60 × 3 cm in NLOS scenarios, which are shown in Figure 4a. Moreover, Figure 4a shows that the two proposed methods outperform the 2D MP method, because the proposed methods concentrate the LOS path energy and suffer from less model mismatch errors, thus achieve higher ‘SINR’ of the LOS path component.

At high SNRs in LOS scenarios, the SNR of the LOS path component is much higher than that in NLOS scenarios due to the dominant LOS path, hence all the three methods perform much better in LOS scenarios than in NLOS scenarios by comparing Figure 4b with Figure 4a. It can be seen from Figure 4b that the performance of the 2D MP algorithm and the FRFT-MP method only improves with virtual antenna array size increasing from 60 × 0.5 cm to 60 × 1.0 cm and then it is worsened with virtual antenna array size increasing from 60 × 1.0 cm to 60 × 3.0 cm, and the FRFT-IFT method performs better with virtual antenna array size increasing from 60 × 0.5 cm to 60 × 2.0 cm and performs worse with virtual antenna array size increasing from 60 × 2.0 cm to 60 × 3.0 cm, because the multipaths’ phase model mismatch errors dominate the performance of the three methods in most virtual antenna array sizes considering the ‘SINR’ of the LOS path component is considerably high in LOS scenarios. Figure 4b also shows that the performance of the proposed algorithms are generally inferior to that of the 2D MP algorithm, especially at cases of relative small virtual sizes, since the performance of the 2D MP algorithm in LOS scenarios is dominated by only the LOS path’s phase model mismatch error, which is greatly less than the multipaths’ phase model mismatch errors in NLOS scenarios. Besides the LOS path’s phase model mismatch error, the proposed FRFT-IFT method and the FRFT-MP method are also affected by the multipath interference and the residual multipath interference, respectively.

Considering that the virtual antenna array size is imprecisely determined and is deliberately chosen as big as allowed to make more of the multipath characteristics, and whether the indoor environment is NLOS or LOS scenarios is unknown, the proposed algorithms, especially the FRFT-MP algorithm, can achieve more robust performance than the 2D MP algorithm by taking the above simulation results into account.

## 5. Conclusions

In this paper, a novel virtual antenna array and fractional Fourier transform-based 2-dimension super-resolution TOA estimation algorithm was presented without requiring a real antenna array. By employing the virtual antenna array idea and modeling the multipath phase variation as a quadratic function, the proposed algorithm can make more use of the multipaths’ characteristics in the space dimension, thus it can enhance the multipath resolution and achieve more robust TOA estimation performance, especially in NLOS scenarios. Simulation results confirm the effectiveness of the proposed algorithm.

## Figures and Tables

**Figure 1 sensors-19-00638-f001:**
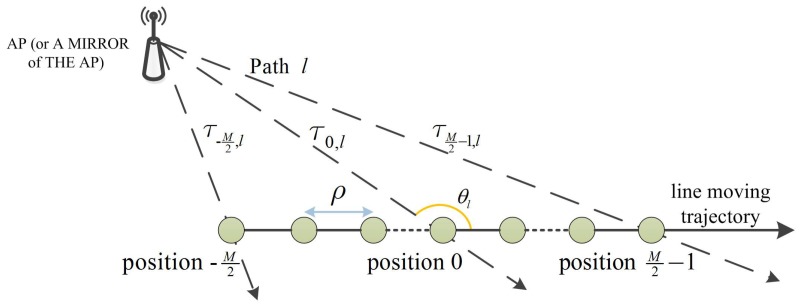
The TOAs of the *l*-th path at equi-spaced positions on a line moving trajectory.

**Figure 2 sensors-19-00638-f002:**
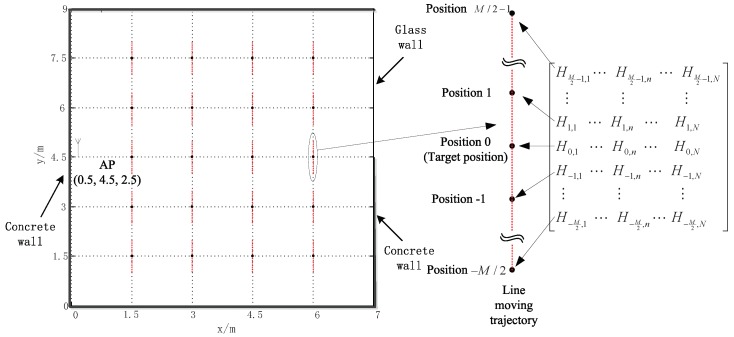
The 2-dimension layout of the indoor WLAN environment.

**Figure 3 sensors-19-00638-f003:**
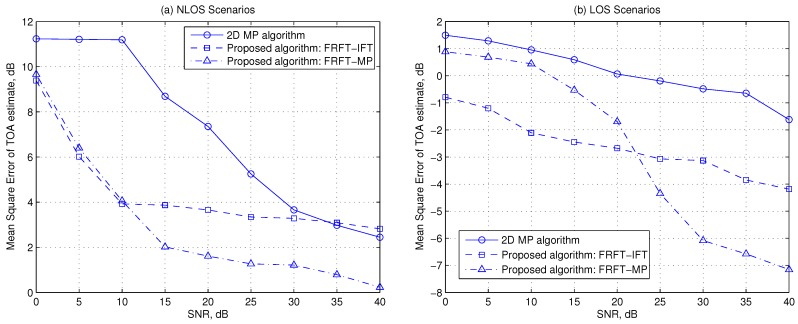
Mean square error versus SNR in NLOS scenarios and LOS scenarios, respectively.

**Figure 4 sensors-19-00638-f004:**
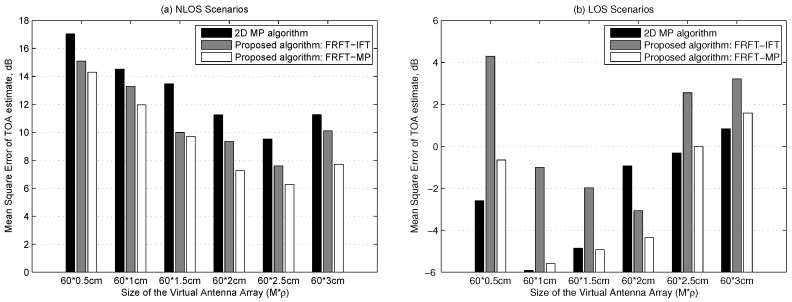
Mean square error versus virtual antenna array size in NLOS scenarios and LOS scenarios, respectively.

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
