# Peer review of "Virtual Antenna Array and Fractional Fourier Transform-Based TOA Estimation for Wireless Positioning"

_sensors, 2019, doi:10.3390/s19030638_

Round 1
Reviewer 1 Report
The paper is well written. It contains only a few shortcomings which should be corrected in the final manuscript:
· Some abbreviations in the text are explained more than once (CFRs, CIR, FRFT, IFFT, MP)
· It would be useful to add some picture to clarify the equation (2).
· It is not clear how the DFRFT has been implemented for the simulations.
· It should be mentioned in the paper abstract that the TOA estimate is derived for an OFDM signal.
· It is not clear how the authors found that the attenuation of a human body at 5 GHz is 15 dB (line 200).
Author Response
The authors would like to thank the anonymous reviewers and the editor for their constructive comments and useful suggestions on our manuscript 412985. In this reply letter, we explain our revisions and provide the point-by-point responses to the comments of the reviewers and the editor. The point-by-point responses to the comments of the reviewers are listed in the attached reply letter. For clarity, the comments of the reviewers are marked blue, and our answers are marked black. Moreover, all the revisions in the revised manuscript are marked black for reviewing.

Reviewer 2 Report
The paper gives the environment, assumptions, requirements and objectives of the problem in hand, and points out major issues or difficulties when dealing with the problem and the system design. However, it can be improved as follows:
1. This paper is based on a number of strong assumptions, which could be critical to the study. It would help if the authors give a detailed description on these assumptions/models .
2. The simulation part can be improved.
Author Response

(The authors gave the same response as above.)

Reviewer 3 Report
This paper proposed a virtual antenna array and fractional Fourier transform (FRFT) based
2-dimension super-resolution time-of-arrival (TOA) estimation algorithm for Wireless
Positioning. The topic sounds interesting. However, in order to further improve this paper’s quality, some minor suggestions for the authors are provided as follows: There are some typos in the paper. The authors should be check again. The results sound interesting in the paper, but the authors may state clearly in experimental assumptions. More channel models and simulation results are needed to show your proposed algorithms’ performance. Fig. 3 showed the performance is very different in both channel cases. The authors may state clearly. The authors may also state clearly their proposed methods and contributions of the paper. In general, I would recommend to be accepted the paper for publication after minor revision.
Author Response

(The authors gave the same response as above.)

Round 2
Reviewer 2 Report
The authors have addressed my concerns in a satisfactory manner.